# The Yin and Yang Effect of the Apelinergic System in Oxidative Stress

**DOI:** 10.3390/ijms24054745

**Published:** 2023-03-01

**Authors:** Benedetta Fibbi, Giada Marroncini, Laura Naldi, Alessandro Peri

**Affiliations:** 1“Pituitary Diseases and Sodium Alterations” Unit, AOU Careggi, 50139 Florence, Italy; 2Endocrinology, Department of Experimental and Clinical Biomedical Sciences “Mario Serio”, University of Florence, 50139 Florence, Italy

**Keywords:** apelin, APJ, apelinergic system, oxidative stress

## Abstract

Apelin is an endogenous ligand for the G protein-coupled receptor APJ and has multiple biological activities in human tissues and organs, including the heart, blood vessels, adipose tissue, central nervous system, lungs, kidneys, and liver. This article reviews the crucial role of apelin in regulating oxidative stress-related processes by promoting prooxidant or antioxidant mechanisms. Following the binding of APJ to different active apelin isoforms and the interaction with several G proteins according to cell types, the apelin/APJ system is able to modulate different intracellular signaling pathways and biological functions, such as vascular tone, platelet aggregation and leukocytes adhesion, myocardial activity, ischemia/reperfusion injury, insulin resistance, inflammation, and cell proliferation and invasion. As a consequence of these multifaceted properties, the role of the apelinergic axis in the pathogenesis of degenerative and proliferative conditions (e.g., Alzheimer’s and Parkinson’s diseases, osteoporosis, and cancer) is currently investigated. In this view, the dual effect of the apelin/APJ system in the regulation of oxidative stress needs to be more extensively clarified, in order to identify new potential strategies and tools able to selectively modulate this axis according to the tissue-specific profile.

## 1. Physiology of the Apelin/APJ System

Apelin is a biologically active neuropeptide that was first isolated in 1998 from bovine stomach extracts [1,2] and identified as the endogenous ligand for the orphan receptor APJ, which was characterized in 1993 as a seven transmembrane G-protein coupled receptor (GPCR) with high affinity (homology of 40–50% in the hydrophobic transmembrane region) with the angiotensin II receptor type 1a [3,4,5].

In humans, apelin is encoded by the APLN gene, which is located on the long arm of X chromosome (Xq25-q26.1) and encodes the 77-aminoacid precursor peptide pre-pro-apelin [2], whose enzymatic hydrolysis originates several active peptide fragments able to activate APJ by their common C-terminal sequence [6]. Apelin isoforms have 12, 13, 15, 16, 17, 19, 28, 31, 36, or 55 aminoacids and display a distinct receptor binding affinity [6], with apelin-13 representing the most effective activator of APJ [7], followed by apelin-17 and apelin-36 [8].

As a GPCR, APJ interacts with G proteins (mainly Gi/o and Gq/11), leading to the modulation of several different signaling pathways after ligand binding. Specifically, via Gi/o, the apelin/APJ system activates the phospho-inositide 3-kinase (PI3K)/AKT (also named protein kinase B, PKB) and protein kinase C (PKC)/extracellular signal-regulated kinase 1/2 (ERK 1/2) pathways, thereby being involved in the regulation of apoptosis, cell proliferation, neuroinflammation, and oxidative stress [9,10]. Moreover, Gi/o is implicated in the downregulation of protein kinase A (PKA) by inhibiting cAMP production [10]. Upregulation of phospholipase C beta (PLCβ) by Gq/11 triggers the generation of diacylglycerol (DAG) and inositol 1,4,5-triphosphate (IP3), which lead to the initiation of the PKC cascade and the intracellular release of Ca^2+^, respectively [10]. Both AKT activation and increase of intracellular Ca^2+^ induces nitric oxide synthase (NOS), thus promoting vasodilation. Binding of apelin to APJ can also result in the autophosphorylation of the receptor through G protein-coupled receptor kinase (GRK). This event initiates a β-arrestin-mediated response involving the desensitization and clathrin-dependent internalization of APJ, which can activate G protein-independent signaling pathways [10,11]. Finally, APJ has also been shown to activate G13 in human umbilical vein endothelial cells, leading to histone deacetylases (HDAC) type 4 and 5 inactivation, activation of myocyte enhancer factor-2 (MEF2) and expression of MEF2 target gene Kruppel-like factor 2 (KLF2) [12] (Figure 1).

APJ and apelin are both highly conserved among species and widely expressed in rodents and human tissues, including lung, heart, spinal cord, brain, placenta, endocrine (thyroid, parathyroid, adrenal, pituitary), gastrointestinal and urinary apparatuses, bone marrow, skeletal and smooth muscles, and adipose tissue, among others [13]. The capability of APJ to interact with several G proteins according to cell types (i.e., heterologous signaling) and stimulate different intracellular pathways explains the variety of biological effects potentially mediated by the apelin/APJ system: vasodilation and lowering of blood pressure [14], increase of cardiac contractility and heart rate [15,16], control of pituitary hormone release, drinking behavior and body fluid homeostasis [17], neuroendocrine stress response [18], food intake and appetite regulation [17,19], glucose metabolism and insulin sensitivity [20], promotion of cell proliferation, migration and angiogenesis, and regulation of gastrointestinal and immune functions [21]. The type and magnitude of downstream events may be cell type- and context-dependent, since apelin isoforms induce different APJ trafficking [22,23]. It is worth noting that although apelin-13 and apelin-36 are able to promote the internalization of APJ, only apelin-13-internalized receptors can be rapidly recovered to the cell surface [24]. Conversely, apelin receptors internalized after binding to apelin-36 represent a target for lysosomal degradation [25].

Elabela is a micropeptide recognized as the second endogenous ligand for APJ [26]. Its involvement in physiological and pathological conditions will not be discussed in this review.

## 2. Apelin/APJ System and Oxidative Stress

Oxidative stress is a condition characterized by an excessive accumulation of reactive oxygen species (ROS) in cells and tissues, which overwhelms the normal dynamic homeostasis and the ability of a biological system to detoxify them. The imbalance between ROS and antioxidants exerts harmful effects on several cellular structures (proteins, lipids, and nucleic acids) [27] and processes (protein phosphorylation, transcriptional factors activation, apoptosis, differentiation, and immunity) [28], thus leading to cell and tissue damage. In this view, oxidative stress is considered as one of the underlying mechanisms of the onset and/or progression of several diseases (i.e., cancer, diabetes, metabolic disorders, atherosclerosis, and cardiovascular diseases) [29].

Mitochondria are the major intracellular site of energy metabolism regulation and therefore they are heavily involved in ROS production [30]. Both enzymatic and non-enzymatic (oxygen reaction with organic compounds, cell exposure to ionizing radiations, mitochondrial respiration) reactions participate in ROS generation from both endogenous (inflammation, ischemia, immune cell activation, infections, cancer, aging) and exogenous (chemical drugs and solvents, smoke, radiations, alcohol) sources [31,32].

As a consequence of apelin/APJ system characterization and evidence of its involvement in the regulation of many intracellular pathways and cell functions, it was not long before there was a demonstration of a close link between this axis and oxidative stress. In fact, not only the myocardial APLN gene expression and protein secretion have been shown to be upregulated by hypoxia via activation of hypoxia-inducible factor (HIF) [33], but the crucial role of apelin in regulating oxidative stress-related processes was also revealed in many tissues and pathological conditions.

Although the activation of apelin/APJ-associated signaling pathways is secondary to both ROS-dependent and ROS-independent stimuli, this review focused on the role of the apelinergic system in oxidative stress-mediated pathologic conditions.

### 2.1. Oxidative Stress-Linked Hypertension, Atherosclerosis, and Pre-Eclampsia

In the vascular system, apelin and APJ are expressed by endothelial and vascular smooth muscle cells (VSMCs), where they are implicated in a complex regulation of blood vessels’ function [34]. Under physiologic conditions, apelin binding to APJ results in vasodilation and transient hypotension by modulating both NO synthesis (via PI3K/Akt and IP3/Ca^2+^ pathways) and the renin–angiotensin–aldosterone system (RAAS). Indeed, its counterregulatory role against angiotensin II-dependent vasopressor stimulation [35,36,37,38] is at least in part secondary to the upregulation of angiotensin converting enzyme 2 (ACE2), which is a negative modulator of RAAS [39,40].

Oxidative stress and vascular NO bioavailability imbalance represent the major etiopathogenetic factors of vascular injury and hypertension [41], with angiotensin II and RAAS acting as crucial triggers of ROS production (e.g., by angiotensin II-induced activity of mitochondrial NADPH oxidase 4, NOX4, which is the upstream signaling molecule of ERK) and endothelial NOS (eNOS) inhibition [42,43,44]. As expected, based on the endothelium-dependent vasodilative properties of apelin, different isoforms of this peptide were demonstrated to mitigate hypertension in in vivo models, with apelin-12 exhibiting the greater effect on blood pressure lowering after intraperitoneal injection of apelin-12, apelin-13, and apelin-36 in anesthetized rats. The absence of a significant antihypertensive effect in APJ-deficient mice suggests that apelin binding to an intact endothelial APJ is required for its vasodilative action [45].

Oxidative stress is the major trigger in the initiation and progression of atherosclerosis. Through the upregulation of selected genes [46], ROS promote mitogenicity and inhibit apoptosis of VSMCs, which contributes to the recruitment of circulant inflammatory cells and the production of extracellular matrix and cytokines, thus participating both in early- and late-stage atherogenesis [47]. Apelin/APJ has been demonstrated to be involved in the development of hypercholesterolemia-associated atherosclerosis similarly to angiotensin II/AT1, which promotes endothelial dysfunction and myosin light chain phosphorylation in VSMCs [48,49,50,51]. By exerting an opposite role on RAAS function to that previously described, apelin-13 is able to activate the ERK-Jagged-1/Notch3-cyclin D1 pathway [52], NOX4 expression and NOX4-derived ROS generation, and oxidative stress-linked proliferation in VSMCs [53]. Accordingly, APJ deficiency can prevent oxidative stress-induced atherosclerosis and protect blood vessels from atherosclerotic plaques [53,54]. In parallel, apelin/APJ-dependent activation of ERK increases the endothelial expression of intercellular adhesion molecule-1 (ICAM-1) and vascular cell adhesion molecule-1 (VCAM-1), and the release of monocyte chemoattractant protein-1 (MCP-1) through the NF-κB/JNK signaling pathway, thus leading to monocyte recruitment and adhesion to endothelial cells [55,56]. Hence, apelin/APJ and oxidative stress seem to be involved in early atherogenesis via the activation of the NOX4-ROS-NF-κB/ERK signaling pathway in VSMCs and in the endothelium.

Abnormal proliferation and migration of VSMCs results in a large number of cells able to penetrate the endothelial layer, deposit in the arterial intima, and secrete bone morphogenetic proteins, which can promote the spontaneous calcification of plaques in late-staged atherosclerosis [57,58,59]. Abnormal apoptosis of mouse aortic vascular smooth muscle cells (MOVAS) secondary to intracellular oxidative stress has been closely related to vascular calcifications through ERK and PI3K/AKT pathways, which both affect MOVAS osteogenic differentiation and calcium deposition [5,60,61]. Zhang et al. have recently reported that apelin-13 significantly reduced high glucose-induced proliferation, invasion, and osteoblastic differentiation of MOVAS—therefore suppressing vascular calcification processes—by inhibiting ROS-mediated DNA damage and regulating ERK and PI3K/AKT pathways [62]. However, apelin’s ability to abrogate the development of atherosclerosis by increasing NO bioavailability and antagonizing angiotensin II cellular signaling was also described [63].

The exact pathophysiological mechanism of pre-eclampsia (PE) is not clearly defined, but abnormal placentation with angiogenic factors levels disproportion and placental insufficiency, increased inflammation, and oxidative stress are known to exert critical roles [64,65]. Adipokines including resistin, adiponectin, and apelin are released even from the placenta during pregnancy [66], and a significant decrease in circulating apelin levels has been demonstrated in PE women compared to normal pregnancies [67,68,69,70]. Circulating apelin decreases in the middle of pregnancy and rises again in the third trimester in healthy pregnancy [71]. Hence, maternal concentrations of apelin lower than expected may play a key role in the etiology of PE. Circulating apelin concentrations showed a significant negative correlation with mean arterial blood pressure, proteinuria, serum soluble fms-like tyrosine kinase-1 (sFlt-1, a soluble form of VEGF/PLGF receptors which acts as an effective scavenger of VEGF and PLGF and sensitizes maternal endothelium to proinflammatory cytokines, thus inducing endothelial dysfunction and multiorgan damage), soluble endoglin (sEng, that acts as a limiting factor for eNOS activity), and IFN-γ levels in PE compared to control women [72]. Furthermore, a positive correlation of apelin levels with serum placental growth factor (PLGF), VEGF and IL-10 levels, and superoxide dismutase (SOD) and catalase activities was also recognized [72]. However, apelin administration significantly improved sFlt-1 and sEng values in the treated group. These results, which are in line with previous reports stating that inflammation is one of the mechanisms of PE by inducing placental ischemia and endothelial dysfunction [73,74], also strengthen the effect of apelin in the pathogenesis of PE.

### 2.2. Oxidative Stress and Diabetic Microvascular Complications

The role of the apelinergic system on the endothelial function accounts for its close association with diabetic microvascular complications, which have, in oxidative stress, one of the underlying pathogenetic mechanisms [75]. In the kidney of diabetic mice, apelin was able to restore antioxidant enzymes’ activity and reduce oxidative stress, thus preventing chronic injury [76] and progression of diabetic nephropathy [77]. Moreover, the evidence of apelin-induced inhibition of ROS generation in an in vitro model of cortical neurons supports the hypothesis of its positive effect in preventing the occurrence of diabetic neuropathy [78]. Nevertheless, the mRNA levels of APJ, apelin, and VEGF are all upregulated in the vascular tissue membrane in proliferative diabetic retinopathy [79], and apelin/APJ was demonstrated to be involved in retinal neoangiogenesis by promoting the expression of VEGF [80,81]. Hence, apelin is supposed to exert a pathogenetic effect in the onset of diabetic retinopathy, and the inhibition of the apelinergic system has been proposed as an effective tool to prevent it.

### 2.3. Oxidative Stress and Cardiac Function

The role of apelin/APJ in myocardial homeostasis and pathology is uncertain and data from literature are conflicting.

On the one hand, it was linked to a protecting effect against ventricular hypertrophy in murine models, where apelin was reported to reduce oxidative stress induced by hydrogen peroxide or 5-hydroxytryptamine [82], and endoplasmic reticulum stress [83]. Similarly, in a model of ischemia-induced heart failure, apelin was proved to reduce ROS production and to ameliorate cardiac dysfunction and RAAS hyperactivation-associated fibrosis, via inhibiting the PI3K/Akt signaling pathway [84].

Peripheral and coronary vasodilatation and improved cardiac output were observed even in patients affected by chronic heart failure after apelin injection [85].

Contrastingly, an increased expression of cardiac myosin and β-MHC (β-myosin heavy chain) mRNA was observed in normotensive rats 15 days after chronic infusion of apelin-13 into the paraventricular nucleus, thus indicating a role of the peptide in the induction of cardiac hypertrophy [86].

### 2.4. Oxidative Stress and Ischemia/Reperfusion Injury

Ischemia/reperfusion (I/R) injury (IRI) consists of the paradoxical exacerbation of cellular dysfunction and death after restoration of blood flow to previously ischemic tissues. Oxidative stress and inflammation secondary to hypoxia-induced production of ROS are the main determinants of cellular and tissue damage [87], which are sustained by activation of matrix metalloproteinase enzymes and degradation of the extracellular matrix and tight junction proteins around endothelial vascular cells [88].

During I/R in in vivo models, apelin is able to protect myocardiocytes against oxidative stress and inhibit mitochondrial oxidative damage and lipid peroxidation by activating eNOS and reperfusion injury salvage kinase (RISK) [89,90]. Hemodynamically, it results in reduced left ventricular preload and afterload, improved cardiac contractility [91], and reduced infarct size [92]. The effect of apelin-13 on post-myocardial infarction repair is partially mediated by an increase of myocardial progenitor cells in the infarcted hearts [93].

Epidemiological data show that diabetes is the most important risk factor for cardiovascular diseases and IRI, with a 2–6-fold increased mortality compared to non-diabetic conditions [94]. Results from animal models showed that heart failure was more severe in diabetic IRI rats compared to non-diabetic IRI rats, and that apelin overexpression significantly decreased injury size and heart weight index and improved cardiac function [95]. Upregulation of PPARα (a well-known modulator of lipid metabolism, antioxidant defense, mitochondrial and endothelial functions, atherosclerosis, and inflammation) and inhibition of apoptosis (enhanced Bcl-2 levels and decreased Bax and cleaved caspase-3 levels) and oxidative stress via the PI3K and p38MAPK pathways has been characterized as the major determinant of apelin’s cardio-protective effects [96,97].

In lungs, IRI often occurs after pulmonary oedema or acute respiratory distress syndrome [98]. Apelin-13 administration to lung IRI rats resulted in a mild damage of alveolar structures, a reduced number of erythrocytes and inflammatory cells, and lower inflammatory cytokines (IL-1β, IL-6 and TNF-α) expression levels [99]. These morphological and molecular changes observed in tissues were associated with an increase of PaO_2_ and a decrease of PaCO_2_ compared to non-apelin-treated IRI rats, thus suggesting that apelin/APJ could minimize IRI by improving lung oxygenation and peroxidation. Finally, apelin-induced expression of uncoupling protein 2 (UCP2), an anionic carrier located on mitochondria which increases SOD activity and improves cell survival in a reduced ROS environment [100], could imply a direct effect of apelin/APJ in ameliorating mitochondrial damage [99].

In the brain, ischemia-induced injury is not only considered as an outcome of inadequate oxygen supply, but it has also been related to an excessive amount of ROS, which lead to cellular and protein dysfunction [101,102,103] and tissue disruption [104]. The degradation of the extracellular matrix secondary to IRI-associated oxidative stress leads to blood–brain barrier (BBB) destruction and vasogenic edema [105], which is a severe consequence of ischemic brain stroke, resulting in a 5% mortality rate [106,107,108]. The subsequent reperfusion contributes to cerebral oedema by initiating the activation of several destructing signaling pathways, including inflammatory responses, alteration of cellular receptors, ion imbalance, oxidative stress, changes in water channel expression, activation of proteinase enzymes, as well as changing tight junction proteins expression [109,110,111,112]. Apelin-13’s ability to significantly decrease brain IRI is mediated by different mechanisms. Gholamzadeh et al. showed that, in mice, oxidative stress markers increased due to ischemia, and that the injection of apelin-13 only 5 min before the onset of reperfusion could significantly reduce vasogenic cerebral oedema and protect BBB integrity [113]. Apelin-13 administration also decreased the expression of endothelin-1 receptor type B [113], whose up-regulation in astrocytes and endothelial cells is associated with metalloproteinase activation [114]. By the activation of ERK1/2 intracellular pathway, the apelinergic system inhibited the production of ROS and increased SOD activity [115]. In parallel, apelin-13 was able to inhibit the ROS-mediated inflammatory response of ischemic stroke by activating the phosphorylation level of AMP-activated protein kinase (AMPK) and the expression of nuclear factor erythroid 2-related factor 2 (Nrf2) [116]. AMPK signaling was also reported to participate in the antiapoptotic role of apelin-13 in ischemic stroke [117]. Conversely, apelin-36-mediated decrease of Bax and caspase-3 levels associated with IRI was related to the PI3K/Akt pathway [118], inhibition of ER stress/unfolded protein response (UPR) activation induced by brain I/R injury [119], and SK1/JNK/caspase-3 apoptotic pathway [120], whereas apelin-12 neuroprotection after ischemia was associated with restrainment of the c-Jun N-terminal kinase (JNK) and p38MAPK signaling pathways of apoptosis-related MAPKs family [121].

Autophagy is a homeostatic process involved in the lysosomal-dependent degradation and elimination of damaged and/or misfolded proteins and organelles. It is negatively modulated by the AMPK/mammalian target of the rapamycin (mTOR) axis [122], and apelin-13 was suggested to attenuate traumatic brain-associated IRI by suppressing autophagy [123].

Finally, apelin/APJ reduced renal IRI by promoting the activity of the mitochondrial enzymes SOD, catalase, and glutathione peroxidase, and decreasing the formation of hydroxyl radicals and malondialdehyde [124].

### 2.5. Oxidative Stress, Obesity, and Insulin Resistance

Data from in vivo obesity models suggest that apelin may function as an adipokine [125,126,127]. Serum levels of this neuropeptide positively correlate with insulin resistance and obesity [125,126,127], and inflammation (particularly by TNF-α production) and oxidative stress have been proposed as the link between apelin/APJ and insulin resistance [128]. In skeletal muscle, apelin enhances the expression of mitochondrial biogenesis markers and enzymes (e.g., citrate synthase, β-hydroxyacyl-CoA dehydrogenase, cytochrome c oxidase) and the content of proteins involved in the assembly of mitochondrial respiratory chain complexes [129,130]. In adipocytes, the apelin/APJ axis prevents the generation of ROS by stimulating the expression of antioxidant enzymes (through MAPK/ERK and AMPK pathways) and inhibiting the expression of pro-oxidant enzymes [131].

The direct effect of insulin on the adipocytic production of apelin is supported by the statistical association among different markers of adiposity, related risk factors, and apelin expression from rat subcutaneous and retroperitoneal adipose tissue [132]. On the other hand, the correlation between apelin mRNA levels and markers of hepatic oxidative stress highlighted a possible role of the apelinergic system in obesity-induced liver oxidative steatosis and dysfunction [132]. Accordingly, exogenous apelin injection restored glucose tolerance and increased glucose utilization in peripheral tissues in high fat diet mice with hyperinsulinemia, hyperglycemia, and obesity [133].

### 2.6. Oxidative Stress and Aging

Apelin has been reported to be downregulated with age in different tissues, and its absence accelerates the onset and progression of aging. Again, oxidative stress is considered to be the link between apelin and the aging process [134]. Specifically, increasing evidence has shown that the apelinergic system participates in autophagy [135,136] and alleviates oxidative stress [82,131,137], which contributes to the development of aging.

### 2.7. Oxidative Stress in the Central Nervous System

Apelin and APJ mRNAs are widely expressed in neuronal cell bodies and fibers throughout the entire central nervous system (CNS), such as in the thalamus, subthalamic nucleus, pituitary gland, hippocampus, basal forebrain, frontal and piriform cortex, striatum, corpus callosum, substantia nigra, olfactory tract, amygdala, central gray matter, spinal cord, and cerebellum [34,138,139].

This broad localization fits with the huge impact of the apelinergic system in neuroprotection, which goes through several mechanisms: suppression of oxidative stress, inhibition of apoptosis and excitotoxicity, and modulation of inflammatory responses and autophagy. Interestingly, these different processes are frequently interconnected and regulated by the same intracellular pathways [45]. Indeed, apelin’s beneficial properties on ethanol-induced memory impairment and neuronal injury of rats are sustained by inhibitory effects on hippocampal oxidative stress, apoptosis, and neuroinflammation [140]. Specifically, the administration of apelin-13 was observed to increase antioxidant enzymes’ activity and glutathione concentration, reduce lipid peroxidation and the number of active caspase-3 positive cells, and attenuate TNF-α production and glial fibrillary acidic protein (GFAP) as a neuroinflammation mediators [140].

The regulatory role of apelin/APJ in neuroinflammation is exerted by suppressing the activity of microglia, astrocytes, and other inflammatory cells [141,142]. Microglia are the innate immune cells of the CNS, able to both eliminate pathogens and cell debris and contribute to neuronal regeneration after tissue damage through the acquisition of different activated phenotypes: M1 cells produce pro-inflammatory cytokines and ROS, causing cytotoxic effects, whereas M2 cells synthetize anti-inflammatory cytokines and stimulate tissue repair [143]. In an in vivo model of ischemic stroke, apelin-13 reduced the expression of pro-inflammatory cytokines and chemokines (IL-1β, TNF-α, macrophage inflammatory protein 1α or MIP-1α, monocyte chemoattractant protein 1 or MCP-1) produced by M1 microglia and increased the expression of the M2-derived anti-inflammatory cytokine IL-10 [144]. The shift of microglial M1 polarization toward the M2 phenotype may be sustained by the blockage of STAT3 signal [145]. The activation of the brain-derived neurotrophic factor (BDNF)-Tyrosine Kinase receptor B (TrkB) signaling pathway, and the inhibition of the NF-κB pathway and endoplasmic reticulum (ER) stress-associated AMPK/TXNIP/NLRP3 inflammasome are other targets of apelin-mediated suppression of neuroinflammation, resulting in improvement of cognitive dysfunction, depressive-like behavior, and early brain injury subarachnoid hemorrhage [142,146,147]. The overactivation of ER stress induced by ROS, with the aim to remove the damaged elements, induces calcium release and reinforces oxidative stress, which promote further microglia activation and leukocyte infiltration into the brain, which subsequently trap in a vicious circle to exacerbate brain injury after stroke [148,149]. In this view, apelin-13 activates AMPK and degradation of TXNIP, which suppresses the overactivation of ER stress and reduces the level of NLRP3 [150].

Excitotoxicity is a complex process of neuronal sufferance and death triggered by the excessive levels of neurotransmitters, which result in a pathologic stimulation of specific receptors. Glutamate neurotoxicity (GNT) is a condition characterized by time-dependent damage of several cell components driven by a massive cell influx of calcium ions and activation of enzymes, including phospholipases, endonucleases, and proteases such as calpain [151,152]. Among the neuroprotective effects of the apelinergic system, the inhibition of excitotoxicity by the activation of pro-survival pathways (i.e., PI3K/Akt and PKC/ERK1/2) and the regulation of N-Methyl-D-aspartic acid (NMDA) receptor activity [153,154,155,156] have also been described.

Patients who have undergone thoracic and abdominal aortic surgery are frequently faced with nerve injury induced by spinal cord ischemia, which is driven by ROS-induced neuronal apoptosis, neuroinflammation, and autophagy [157]. The intraperitoneal injection of apelin-13 exerted spinal cord protection and recovery of motor function in rats by suppressing autophagy, oxidative stress, and mitochondrial dysfunction [158].

Recent evidence suggests that GnRH neurons are targets of apelin-associated neuroprotection. APJ signaling pathway activation via either apelin-13 or transient overexpression is able to increase GnRH neurons proliferation after H_2_O_2_ exposure and hypoxia, and to stimulate the conversion of G0/G1 to S phase through AKT and ERK-1/2 kinase pathways activation [159]. Therefore, the expression and activation of the apelin/APJ system in GnRH neurons might support a protective mechanism against oxidative stress-induced cell death. Furthermore, the observation of a promoting effect of the apelinergic system on GnRH release in embryonic stem cell-derived GnRH neurons supports the hypothesis of its pro-differentiating role during developmental stages [159].

The antioxidative stress effects of apelin/APJ prompted the research to evaluate its potential correlation with neurodegenerative diseases.

Alzheimer’s disease (AD) is the most prevalent form of dementia in the elderly, characterized by intracellular neurofibrillary tangles (NFTs) and extracellular amyloid beta (Aβ) protein deposits that contribute to senile plaques and progressive neurodegeneration [160]. The neuronal loss that appears in the cerebral cortex and in the hippocampus as a consequence of mitochondrial dysfunction and ROS production is an early event in AD and anticipates senile plaques appearance [161,162]. By activating glycogen synthase kinase-3 (GSK-3) and c-Jun N-terminal kinase (JNK)/p38MAPK, oxidative stress induces Tau phosphorylation and beta-site amyloid precursor protein cleaving enzyme 1 (BACE1) expression, and therefore promotes the production of NFTs and Aβ [163,164,165]. Moreover, dysregulation of intracellular calcium exerts a crucial role in the regulation of familial Alzheimer’s proteins (PSEN1 and PSEN2) and Aβ, which results in altered calcium signaling, loss of synapses, and memory impairment [166]. In this scenario, serum apelin-13 has been shown to be lower in AD patients compared to control subjects [167] and its exogenous administration attenuates Aβ-induced memory deficit in Aβ-treated animals [168]. Subsequent molecular studies revealed that the apelinergic system participates in the pathophysiology of AD via regulating Tau and Aβ [146]. Again, the intracellular mechanisms involved in this complex regulation are multiple: (i) activation of PI3K/AKT phosphorylates and inactivates GSK3β, thus suppressing Tau hyperphosphorylation and Aβ accumulation [169]; (ii) inhibition of Aβ-induced autophagy through mTOR signaling pathway [168]; (iii) inhibition of the synthesis of inflammatory mediators, especially TNF-α and IL-1β [170]; (iv) improvement of cell survival and inhibition of neuronal apoptosis through reduction of cytochrome c, increase of caspase-3, and suppression of intracellular calcium release [170,171]; (v) modulation of excitotoxicity [153]. The effects of the apelinergic system on multiple mechanisms involved in AD pathogenesis make apelin a potential therapeutic agent in AD.

In Parkinson’s Disease (PD), the progressive loss of dopaminergic neurons in the substantia nigra is secondary to the accumulation of misfolded α-synuclein (α-Syn) in cytoplasmic inclusions named Lewy bodies [172,173]. Dysfunction of parkin, a key part of a multiprotein E3 ubiquitin ligase complex which destroys malformed proteins in neurons, is associated with the pathogenesis of PD [174], which is sustained by oxidative stress, microglia activation, and excessive neuroinflammation [175]. The dysregulation of PI3K/Akt and MAPKs cascades is implicated in the imbalance between cellular anti-apoptotic and pro-apoptotic pathways [175]. As in AD, apelin/APJ axis activation was linked to inhibition of apoptosis and dopaminergic neuronal loss, activation of antioxidants and autophagy, prevention of excessive neuroinflammation, suppression of endoplasmic reticulum stress, and glutamate-induced excitotoxicity. In in vitro models of SH-SY5Y cells, apelin-13 pre-treatment preserved the mitochondrial membrane potential, inhibited the release of cytochrome c and cleaved-caspase 3, and reduced ROS production, thus improving cell viability via PI3K-induced Akt activation [8,22]. AMPK/mTOR-dependent activation of autophagy [22] and ERK1/2-mediated attenuation of ER stress [176] contribute to apelin-13 protection against dopaminergic neurodegeneration. Downregulation of ROS and prevention of SH-SY5Y apoptosis was also described for apelin-36 [120]. In agreement with in vitro observations, different in vivo studies confirmed the neuroprotective role of apelin isoforms. Apelin-36 was able to prevent dopamine depletion in the striatum, at least partially via improving antioxidant cellular mechanisms (including SOD and glutathione) and downregulating inducible NOS and nitrated α-Syn expression [120], whereas apelin-13 markedly improved cognitive impairments in 6-OHDA-treated animals [177].

### 2.8. Oxidative Stress and Osteoporosis

High levels of oxidative stress and mitochondrial dysfunction are key regulators of bone marrow mesenchymal stem cells (BMSCs) survival and bone formation [178]. ROS overproduction associated with aging and estrogen deficiency determines the establishment of a “pro-osteoporotic” microenvironment, which alters the commitment of BMSCs and shifts their differentiation from the osteogenic to the adipogenic line [179]. Furthermore, intracellular ROS accumulation promotes BMSCs apoptosis [180,181], induces loss of function and apoptosis in osteoblasts [182], and increases osteoclastic bone resorption [183], thus contributing to the development of osteoporosis. Upon mitochondrial damage, mitophagy (a unique form of autophagy) selectively removes damaged mitochondria and prevents their accumulation and oxidative stress aggravation [184]. Hence, its activation in BMSCs contributes to promoting osteogenic function at the expense of adipogenic commitment [185,186,187,188,189,190]. As expected, based on the essential role of adipokines in bone homeostasis, the apelin/APJ system is a potential therapeutic tool in the treatment of osteoporosis. Endogenous apelin is highly expressed during osteogenesis in human BMSCs [191], whereas both apelin and APJ are downregulated in distal femurs of ovariectomy-induced osteoporotic rats [192]. Accordingly, serum apelin-13 in osteoporotic patients was significantly lower than in osteopenia and normal subjects [193]. Molecular and cellular studies demonstrated that apelin is able to stimulate proliferation and to suppress apoptosis of the osteoblastic cell line MC3T3-E1 [194] and to prevent mitochondrial ROS accumulation [195] and mitophagy [192] in BMSCs via the AMPK pathway [196].

### 2.9. Drug-Induced Oxidative Stress

Cisplatin, a broad-spectrum chemotherapeutic drug which affects DNA replication and inhibits cell division, is burdened by cardio- and ototoxicity [197,198]. Oxidative stress-dependent apoptosis of cardiomyocytes, which are limitedly able to regenerate, results in irreversible cisplatin-induced cardiomyopathy [199,200]. Oxidation resistance is recognized as a key cellular event in the protective effects of apelin-13 in cisplatin-exposed cardiomyocytes, where it efficaciously blocks the mitochondrial apoptosis pathway by inhibiting ROS-mediated DNA damage and p53 phosphorylation and regulating MAPKs and AKT pathways [201]. In the cochlea, excessive ROS production and mitochondrial dysfunction induced by cisplatin are key contributors of cochlear hair cells (HCs) [202,203,204]. Downregulation of apelin expression has been related to cisplatin-induced damage to HCs, and exogenous apelin’s otoprotective effect against cisplatin-induced injury is closely associated with its ability to inhibit ROS production and mitochondrial dysfunction, which are known to potentiate cisplatin-induced apoptosis, via deregulation of JNK signaling [205]. Most recently, apelin-13 administration was demonstrated to reduce nephrotoxicity induced by cisplatin by triggering oxidative stress and inflammation [206].

Bupivacaine is a commonly used local anesthetic which may cause cardiotoxicity via inhibition of PI3K/AKT signaling [207], respiratory chain complexes I, III, and IV [208], and carnitine palmitoyl transferase [209]. As a result, cardiac energy metabolism is altered, and cardiac arrest may occur. In a rat model, apelin-13 treatment reduced bupivacaine-induced oxidative stress, attenuated mitochondrial morphological change and DNA damage, and enhanced mitochondrial energy metabolism through modulation of AMPK cascade, ultimately reversing bupivacaine-induced cardiotoxicity [210].

### 2.10. Oxidative Stress and Cancer

Cancer cells show a great ability to adapt their functions to perturbation of cellular homeostasis, particularly the imbalanced redox status secondary to local hypoxia and high metabolism. The theory of ROS rheostat predicts a fine regulation of ROS production and scavenging pathways to potentiate the antioxidant capacity of neoplastic cells and allow oxidative stress levels compatible with intracellular activities, even if higher than in normal cells [211]. Accordingly, an increased expression of ROS scavengers and low oxidative stress levels were described as crucial for the survival of pre-neoplastic foci in breast and liver cancer stem cells [212,213]. Indeed, oxidative stress is involved in the regulation of several cell functions, which are deregulated in cancer (i.e., cell growth, excitability, cytoskeleton remodeling and migration, autophagy, exocytosis and endocytosis, hormone signaling, necrosis, and apoptosis) [214,215], in the promotion of genomic instability and/or transcriptional errors [216], and in the activation of pro-survival and pro-metastatic pathways [215]. Consequently, the three steps of carcinogenesis (initiation, promotion, progression), local invasiveness and metastatization, and the resistance to treatment are strongly conditioned by the imbalance between ROS and antioxidant production [217]. As strong inducers of ROS generation, chemotherapy and radiotherapy are often unable to definitivly cure cancer: antineoplastic drugs and radiations may eliminate the bulk of cancer cells, but the upregulation of antioxidants in the presence of high ROS levels and ROS-dependent accumulation of DNA mutations are mechanisms that spare cancer stem cells and lead to therapeutic failure [211]. In this very complex scenario, antioxidant inhibitors (e.g., glutathione, HSP90, thioredoxin, enzyme poly-ADP-ribose polymerase or PARP) are considered a promising therapeutic tool in cancer treatment in association with radiotherapy or chemotherapy [211].

In the last 15 years, the role of the apelinergic system in tumorigenesis and cancer progression emerged from several studies and it has been proposed as a novel therapeutic target for different malignant tumors [218]. The apelin/APJ axis is upregulated in glioblastoma, esophageal squamous cell carcinoma, cholangiocarcinoma, and lymphoma, and it has been associated with carcinogenesis [218,219,220,221]. Furthermore, serum apelin levels were correlated with shorter survival, higher incidence of cancer recurrence and resistance to anticancer drugs in some human solid tumors, such as gastric cancer, lung adenocarcinoma, and breast cancer [222].

Hypoxia caused by the hypermorphosis of tumor cells was shown to promote apelin expression [223] via increased ROS-dependent hypoxia inducible factors (HIFs) activation [224], even in cancer stem cells [225]. The promoting effect of apelin/APJ in oxidative stress-associated cancer proliferation was reported in gastric adenocarcinoma cells [160] and melanoma [219], where apelin stimulated cancer cells survival and accelerated tumor growth in addition to allowing intratumoral lymphatic capillary and lymphnode metastatization. In several cancers, apelin may also protect cancer cells from apoptosis [226,227] and may play a role in mediating differentiation of mesenchymal stem cells into cancer stem cells, whose self-renewal is facilitated by activating signaling pathways such as wnt/β-catenin and Jagged/Notch [222]. In breast cancer, increased apelin levels were found to be an independent predictor of HER-2/neu expression and breast cancer phenotype, which accounts for 30% of breast carcinomas and is associated with a more aggressive tumor behavior [228].

The role of apelin/APJ signaling in angiogenesis is also well recognized in different cancers [222]. Growing evidence has suggested that apelin induces the maturation of tumor blood capillaries [229] and stimulates the proliferation of smooth muscle cells by modifying cyclin D1 expression and favoring the progression of cell cycle [230].

## 3. Conclusions

The apelin/APJ system may exert opposite effects on oxidative stress-mediated processes in different tissues and pathologic conditions (Table 1) by promoting prooxidant or antioxidant mechanisms (Figure 2). These contradictory functions, which can be explained by the existence of multiple isoforms of apelin, the activation of different APJ-coupled G proteins and signaling pathways, and context-dependent APJ trafficking, make the apelinergic axis a double-edged sword in regulating oxidative stress-associated diseases. In this view, a full comprehension of the complex role of apelin/APJ in ROS-related physiologic and pathologic processes is crucial, as well as to identify innovative therapeutic tools based on APJ inhibition or activation.

## Figures and Tables

**Figure 1 ijms-24-04745-f001:**
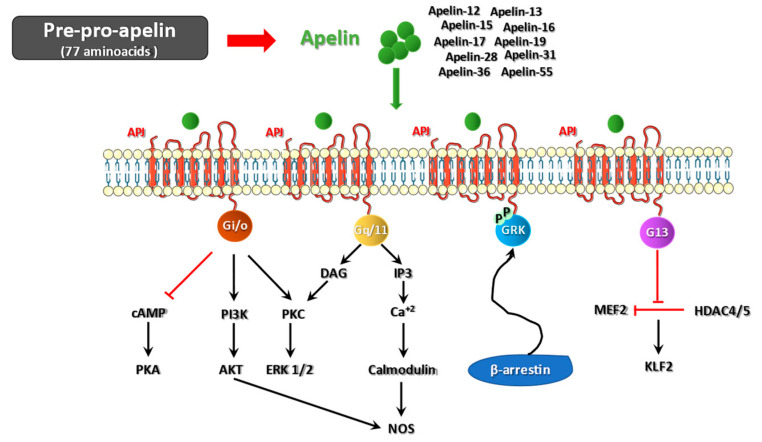
Intracellular pathways modulated by the apelin/APJ system. The 77-aminoacid precursor peptide, pre-pro-apelin, is cleaved in active fragments (apelin-12, apelin-13, apelin-15, apelin-16, apelin-17, apelin-19, apelin-28, apelin-31, apelin-36, apelin-55), which bind the apelin receptor APJ. By interacting with G proteins, apelin/APJ is able to modulate different signaling pathways: inhibition of cAMP generation and protein kinase A (PKA) and activation of phospho-inositide 3-kinase (PI3K)/AKT through Gi/o; activation of protein kinase C (PKC)-dependent extracellular signal-regulated kinase 1/2 (ERK1/2) through Gi/o or Gq/11; initiation of the intracellular release of Ca^2+^ by Gq/11 and inositol 1,4,5-triphosphate (IP3) synthesis; autophosphorylation of APJ through G protein-coupled receptor kinase (GRK) and initiation of the β-arrestin-mediated internalization of the receptor; activation of G13 and inactivation of histone deacetylases (HDAC) type 4 and 5, determining the activation of myocyte enhancer factor-2 (MEF2) and expression of MEF2 target gene Kruppel-like factor 2 (KLF2). Both AKT activation and increase of intracellular Ca^2+^ induces nitric oxide synthase (NOS).

**Figure 2 ijms-24-04745-f002:**
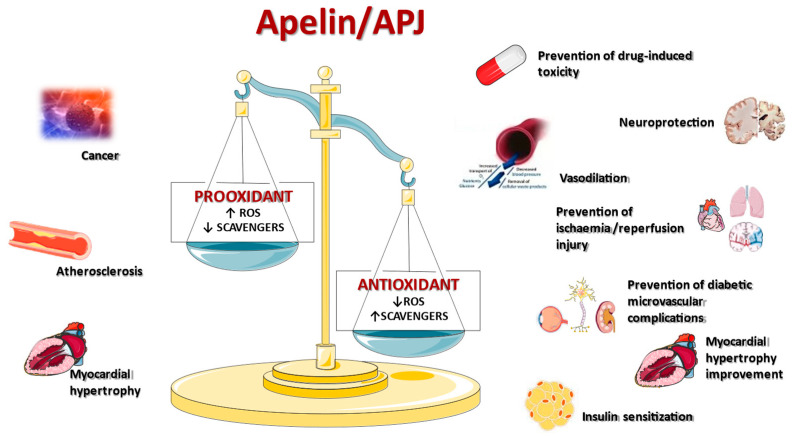
Prooxidant and antioxidant functions of the apelin/APJ system.

**Table 1 ijms-24-04745-t001:** Biological actions of the apelinergic system oxidative stress-related diseases.

Oxidative Stress-Related Effects the Apelinergic System in Different Organs and Tissues	
**Cardiovascular system**	
Vasodilation and blood pressure lowering [35,36,37,38,39,40,45]	- Induction of NO synthesis - RAAS modulation by counterregulating angiotensin II-dependent vasopressor stimulation by upregulating ACE2
Promotion of early atherogenesis [48,49,50,51,52,53,54,55,56]	- Promotion of endothelial dysfunction and myosin light chain phosphorylation in VSMCs - Activation of oxidative stress-linked proliferation in VSMCs - Induction of endothelial expression of ICAM-1 and VCAM-1 and release of MCP-1
Suppression of vascular calcification processes [62,63]	- Reduction of high glucose-induced proliferation, invasion, and osteoblastic differentiation of MOVAS - Increase of NO bioavailability
Prevention of diabetic microvascular complications [75,76,77,78,79,80,81]	- Inhibition of oxidative stress in kidney and neurons - Inhibition of retinal neoangiogenesis
Improvement of cardiac function [82,83,84,85]	- Inhibition of RAAS hyperactivation-associated fibrosis - Coronary vasodilation
Protection of myocardiocytes against IRI and reduction of infarct size in diabetic and non-diabetic patients [89,90,91,92,93,95,96,97]	- Inhibition of oxidative stress - Increase of myocardial progenitor cells in the infarcted hearts - Upregulation of PPARα - Inhibition of apoptosis
Protection of myocardiocytes against cisplatin-induced injury [201]	- Inhibition of mitochondrial apoptosis
Protection of myocardiocytes against bupivacaine-induced injury [210]	- Prevention of DNA damage - Enhanced mitochondrial energy metabolism
**Lung**	
Restrainment of IRI-associated damage after pulmonary oedema or acute respiratory distress syndrome [99,100]	- Reduction of inflammatory infiltrates and proinflammatory cytokines secretion - Upregulation of UCP2 and SOD activity - Improvement of cell survival
**Placenta**	
Low apelin levels correlate with the etiopathogenesis of pre-eclampsia by inducing placental ischemia and endothelial dysfunction [67,68,69,70,71,72]	- Reduced synthesis of angiogenetic factors - Induction of a pro-inflammatory microenvironment
**Central nervous system**	
Protection of neurons and BBB against IRI [113,114,115,116,117,118,119,120,121,122,123]	- Reduction of vasogenic cerebral oedema - Protection of BBB integrity - Inhibition of inflammatory response - Inhibition of apoptosis - Inhibition of autophagy (traumatic brain-associated IRI)
Neuroprotection [45,140,141,142,143,144,145,146,147,148,149,150,151,152,153,154,155,156,157,158,159,160,161,162,163,164,165,166,167,168,169,170,171,172,173,174,175,176,177]	- Inhibition of apoptosis - Inhibition of excitotoxicity - Inhibition of neuroinflammation - Inhibition of autophagy
**Kidney**	
Protection of renal cells against IRI [124]	- Induction of the activity of the mitochondrial enzymes SOD, catalase, and glutathione peroxidase
Protection of renal cells against cisplatin-induced toxicity [206]	- Inhibition of inflammatory response
**Adipose tissue, skeletal muscle, and liver**	
Amelioration of insulin sensitivity [125,126,127,128,129,130,131,132,133]	- Increased expression of mitochondrial biogenesis markers and enzymes - Increased glucose utilization - Reduced liver steatosis and dysfunction
**Bone**	
Maintenance of bone health [192,194,195,196]	- Stimulation of osteoblast proliferation - Suppression of osteoblast apoptosis - Prevention of mitophagy in BMSCs
**Inner ear**	
Protection of cochlear cells against cisplatin-induced injury [205]	- Inhibition of cochlear cells apoptosis
**Cancer cells**	
Increased serum apelin levels correlate with shorter survival, higher incidence of cancer recurrence, and resistance to anticancer drugs [160,218,219,220,221,222,223,224,225,226,227,228,229,230]	- Increased cell proliferation - Increased cell survival and inhibition of cell apoptosis - Increased intratumoral lymphatic capillaries and lymph nodes metastasization - Differentiation of mesenchymal stem cells into cancer stem cells - Maintenance of cancer stem cells self-renewal - Increased angiogenesis

NO: nitric oxide; RAAS: renin–angiotensin–aldosterone system; ACE2: angiotensin converting enzyme 2; VSMCs: vascular smooth muscle cells; ICAM-1: intercellular adhesion molecule-1; VCAM-1: vascular cell adhesion molecule-1; MCP-1: monocyte chemoattractant protein-1; MOVAS: mouse aortic vascular smooth muscle cells; IRI: ischemia/reperfusion injury; UCP2: uncoupling protein 2; SOD: superoxide dismutase; BBB: blood–brain barrier; BMSCs: bone marrow stromal cells.

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
