# Peer review of "The Yin and Yang Effect of the Apelinergic System in Oxidative Stress"

_ijms, 2023, doi:10.3390/ijms24054745_

Round 1

Reviewer 1 Report

The review is written at a very high scientific level. It addresses the controversial role of the apelinergic system, expressed in many tissues and systems and discovered relatively recently. The authors review the physiological role of this system and its involvement in the pathogenesis of many diseases of the central nervous system, cardiovascular system, in metabolic syndrome with hyperinsulinemia and obesity, etc. They elucidate the crucial role of oxidative stress in the pathogenesis of these diseases, explicitly highlighting the modulating effects of apelin in these pathologies.

Author Response

We thank the reviewer for appreciating our work.

Reviewer 2 Report

The activity of apelin, as suggested in the article, has a pleiotropic activity, however there is an extensive review of oxidative stress and the role that regulation of oxidative stress could play, the article seems more like a brief review of diseases chronic degenerative diseases and the participation of oxidative stress in them, although as he mentions there is still much to study the role of apelin as a regulator, it would be worth going deeper either in each of the sections of the diseases that he exposes or making a table, specifically explaining the role of apelina in each of them, supported by the literature, as well as pointing it out.

Author Response

We thank the reviewer for his/her advices. We added a table with specific effects of apelin/APJ system in different organs and tissues, in order to better clarify the multifaceted role of this axis in human diseases.

Reviewer 3 Report

The subjected paper is an excellent overview of the pathological and pharmacological mechanisms in which the peptide apelin and its receptor have a role.  Specific emphasis is put on potential steps involving reactive oxygen species. It has been written by professional endocrinologists, whose activity, however,  has  been exerted in different areas until now.  

The overview of the literature is correct, almost fully exhaustive. The organization of the material in chapters is logical. Language, style are excellent (with exceptions of a few overcomplicated sentences), understanding is facilitated by simple Figures.

What is problematic that only part of the published literature feels essential to suppose a reactive oxygen species step in the mechanism of action of this extracellular signal peptide. Main effects of the  PKA, PI3K, Akt, ERK1/2, calmodulin signal pathways (correctly marked on Fig 1.) are working without the  essential contribution of the ROS system. Most of the specialists will find incorrect the arrows converging toward NOS (?) here. In the paper  it is not clearly separated, what are the proven signal pathways (with ROS involvement or without it), and where a positive proof is existing for the importance of ROS involvement in the pathological and pharmacological  action.

Some specific criticism

Some relevant new literature:

Balci CN et al Annals Anat 2023;246:152027. Apelin and apelin receptor expression and function during implantation.

Gao Y et al. Front Endocrinol2023;13:1076951 No apelin plasma level elevations in polycystic ovary disease.

Liu Q et al. Life Sci 2020;260:118310. Apelin receptor- target in PCOS.

Sharma M et al. Curr Drug Targets 2022;23:1304. Elabela peptide, second endogenous ligand of the apelin receptor.

Topcu A et al. Drug Chem Toxicol 2023;46:77  AP13 administration decreased cisplatin toxicity.

Lines 126-128, Correct the sentence.

Lines 171-178, Too lengthy sentence, hard to perceive.

Line 203 … to reduce   … endoplasmic reticulum?....

Line 230, Correct the comparison.

Line 303  …downregulated… where? In which tissue? Blood plasma?

Author Response

We thank the reviewer for his appreciation and advices.

We specified in the text that the effects of the apelinergic system go beyond oxidative stress-mediated processes, but that we decided to focus our discussion on this target (according to the topic of the special issue, i.e. "Cellular and Molecular Mechanisms in Oxidative Stress-Related Diseases 2.0"). Anyway, ROS-dependent and ROS- independent apelin/APJ actions in cellular biology cannot be clearly separated, yet interactions exist at different levels.

With regard to the comment on Fig. 1 ("arrows converging toward NOS"), the association between apelin-mediated signaling pathways and NOS is known and it has been addressed for instance in the following reference: "Kang, Y.; Kim, J.; Anderson, J.P.; Wu, J.; Gleim, S.R.; Kundu, R.K.; McLean, D.L.; Kim, J. D.; Park, H.; Jin, S.W.; Hwa, J.; Quertermous, T.; Chun, H.J.  Apelin-APJ signaling is a critical regulator of endothelial MEF2 activation in cardiovascular development. Circ Res 2013, 113, 22–31" (n. 12).

We added interesting references and corrected sentences, as suggested. 

Round 2

Reviewer 2 Report

I agree with the corrections